# Rational construction of controllable autoimmune diabetes model depicting clinical features

**Fan Yu**[1], **Xian Zhou**[1], **Xiang Jin**[1], **Shushu Zhao**[1], **Gan Zhao**[1], **Sheng Jiang**[1], **Shuang Geng**[1,2,3]*, **Bin Wang**[1,2,4]*

**1** Key Laboratory of Medical Molecular Virology of MOH and MOE, School of Basic Medical Sciences, Fudan University, Shanghai, China, **2** National Clinical Research Center for Aging and Medicine, Huashan Hospital, Fudan University, Shanghai, China, **3** Department of Microbiology, Immunology and Infectious Diseases, Snyder Institute, University of Calgary, Calgary, Alberta, Canada, **4** Children's Hospital, Fudan University, Shanghai, China

\* crowgs@gmail.com (SG); bwang3@fudan.edu.cn (BW)

**Data Availability Statement:** All relevant data are within the paper and its Supporting Information files.

**Funding:** B.W. is supported by the National Natural Science Foundation of China Key Project

## Abstract

Through animal models, particularly non-obesity diabetes model (NOD), pathological understandings of human autoimmune diabetes have been gained. However, features of those mouse models and the human disease are not sufficiently analogous; it is therefore not unexpected that interventions based on the mouse data fail at an alarming rate in clinical settings. An improvised model that maximally resembles the real pathological course is highly desirable. Here we devised a 'double-hit' strategy, pancreas was first hit by chemical damage (streptozotocin, STZ) to unleash auto-antigens, then hit second time by transient immune-inflammation (regulatory T cell depletion). Comparing to NOD model, this strategy not only induced classical diabetic symptoms, but also depicted the crucial pathogenic features absent in conventional models, such as CD8[+] T cell dominant infiltrates, strong ketoacidosis and epitope-specific T cell responses. In addition, this model allowed synchronized control of disease onset, permitting more refined temporal analysis of disease progression. We believe that this model would yield research outcomes with clinically relevant prediction power unattainable previously.

## Introduction

Type 1 diabetes (T1D), also known as insulin dependent diabetes mellitus (IDDM), is mostly an autoimmune diabetes caused by destruction of insulin-producing pancreatic β cell [1, 2]. Once majority of functional β cell mass is lost, lifelong insulin replacement therapy is required to control hyperglycemia. Patients are often accompanied by fatal ketoacidosis, and complications such as blindness and renal disease often result over time [3, 4].

To understand its autoimmune pathogenesis, animal models were built to depict clinical symptoms. For example, non-obese diabetic (NOD) mice were discovered in 1980s [5], which spontaneously develop autoimmune diabetes, and have been extensively explored through

(31430027), General Program (81672016) and
Major Project (81991492).

**Competing interests:** NO authors have competing
interests.

decades [6]. However, despite monumental mechanistic knowledge has been extracted, few immune-therapeutical discoveries were successfully translated to T1D therapy [7], suggesting limited recreation of human pathogenesis in current models. For instance, islet-specific CD8+ T cells are dominant infiltrations in patients, whereas CD4+ T cell infiltrates are dominant in NOD murine model [8]. Also, fatal ketoacidosis is often observed in patients as well as in dog models, but it occurs only mildly in NOD model [9]. Because of these key differences, "successes" in the NOD mice are intrinsically poor indicators of clinical outcomes. Thus, taking all above frailties into consideration, a new animal model with better analogy to human disease is in demand.

Here we reasoned that, occurrence of autoimmunity is a temporal-spatial alignment of damaged pancreas and activated immunity. The former occurs under certain environmental insults, such as chemically toxicities, viral infections, and environmentally pollutions [10, 11]. Consequently, damaged pancreas releases auto-antigens such as pro-insulin, GAD65, IA2, etc. The latter occurs in response to the stress and newly exposed antigens, an active T cell immunity is awaken which accelerates T1D [12]. By addressing when and where these two aspects were aligned *in vivo*, one may reconstruct with a greater extent of fidelity the human autoimmune diabetes.

This rationale has inspired us to propose a "double-hit" hypothesis, that chemical damage to pancreas, followed by systematic immune activation, may recreate T1D pathogenesis. To test this hypothesis, here the first hit (pancreas damage) was generated by multiple low-dose STZ, and the second hit (systematic immune activation) was accomplished with regulatory T cell depletion. In this report, we compare the pathologies of this new model against those of the human disease.

## Results

### Establishment of double-hit autoimmune diabetes

In line with double-hit rationale, we chose multiple low-dose streptozotocin (STZ) to induce pancreas specific damage [13–15], and diphtheria toxin (DT) dependent Treg depletion to induce transient systematic immune activation [16, 17] (**Fig 1A**). As the result, mice insulted by STZ+DT showed hyperglycemia within 2 weeks, this is completely absent in the lone treatment of STZ (the 1st hit) or DT (the 2nd hit) (**Fig 1B**). Consistently, the STZ+DT mice showed significant β cell destruction as early as day 6, whereas no sign of β cell mass reduction in either STZ or DT alone treatment (**Fig 1C and 1E**). Rapid loss of all islet β cells (day 6 to day 20) suggested strong autoimmunity in STZ+DT group. Slower β cell loss was observed in NOD mice, from prediabetic (10-week of age) to diabetic (17-week of age) (**Fig 1C, 1D and 1E**). Thus, these results showed that double-hit of DEREG mice by STZ+DT insults induced a fast onset diabetic hyperglycemia and β cell destruction.

### Diabetes symptoms in STZ+DT model

To validate this double-hit model, diabetic symptoms were compared with NOD mice. Blood glucose of STZ+DT group rose above 16.7 mmol/L, and diabetes onset rate reached 100% within 2 weeks (**Fig 2A,** left panel, and **Fig 2B**, left panel), showing reasonably good synchronicity. In comparison, NOD mice developed hyperglycemia in several waves, ranging from 13 weeks to 28 weeks of age (**Fig 2A and 2B,** right panels), roughly categorizing into acute and progressive diabetes [18]. Next, abnormal glucose tolerance indicates pre-diabetic progression [19]. Therefore, oral glucose tolerance test (OGTT) was performed on fasted STZ+DT mice at day 6. Blood glucose increased significantly comparing with PBS control mice, confirming impaired islet function in the pre-diabetic phase (**Fig 2C,** left panel). As a comparison, OGTT

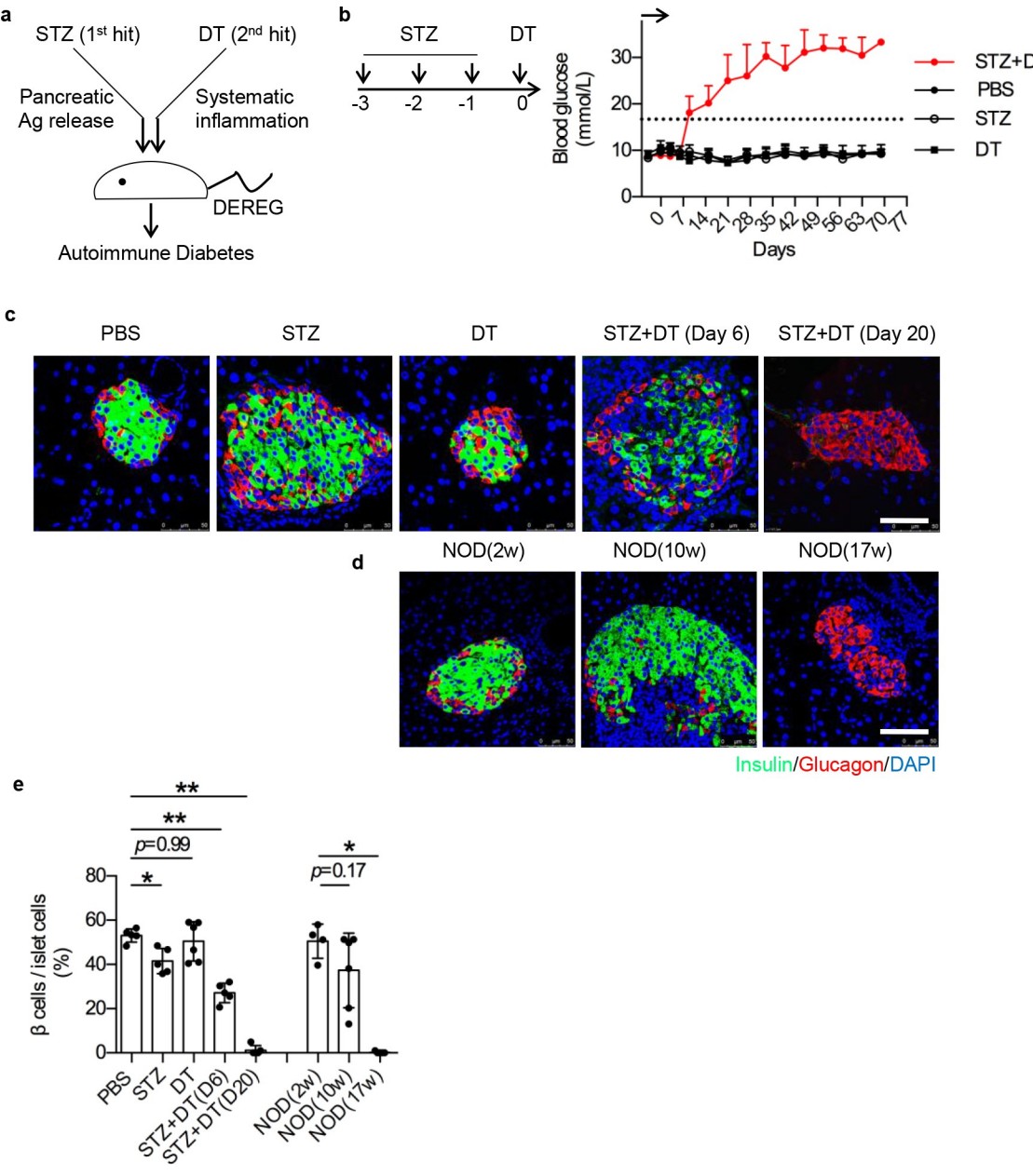

**Fig 1. Establishment of double-hit model by STZ+DT. a**. Rationale of double-hit model. Pancreas was first chemically damaged by multiple low-dose STZ to unleash auto-antigens, then was hit by systematic inflammation (Treg depletion by DT in DEREG mice). DEREG, DEpletion of REGulatory T cells. **b**. STZ and DT administrations. Left, scheme of model setup. Multiple low-dose STZ (50mg/kg) was *i.p.* injected to DEREG mice at day -3, -2 and -1. DT (0.25μg) was *i.p.* injected at day 0. Right, blood glucose fluctuation. PBS was given as control. Arrow represents model setup from day -3 to day 0. Dash line represents blood glucose at 16.7 mmol/L. n = 7. N = 5. **c-d**. Immunofluorescence staining of insulin (green), glucagon (red) and DAPI (blue) in pancreatic islets. PBS, STZ and DT group were analyzed at day 6. STZ+DT group were analyzed at day 6 and day 20 (**c**). NOD mice were analyzed at newborn (2-week of age, 2w), prediabetic phase (10-week of age, 10w) and diabetic phase (17-week of age, 17w) (**d**). Scale bars represent 50μm. **e**. Statistical analysis of β cell loss in terms of insulin+ cell percentage per islet in c and d. n = 5. N = 3. *, $p < 0.05$; **, $p < 0.01$.

of pre-diabetic 10-week NOD mice showed significant increase in blood glucose comparing with 2-week healthy ones (**Fig 2C**, right panel). Furthermore, sérum C-peptide level reliably reflects β cell function [20]. Fasted STZ+DT mice showed reduced C-peptide level, confirming exhausted β cell and insulin insufficiency (**Fig 2D**). Accumulative blood glucose marker

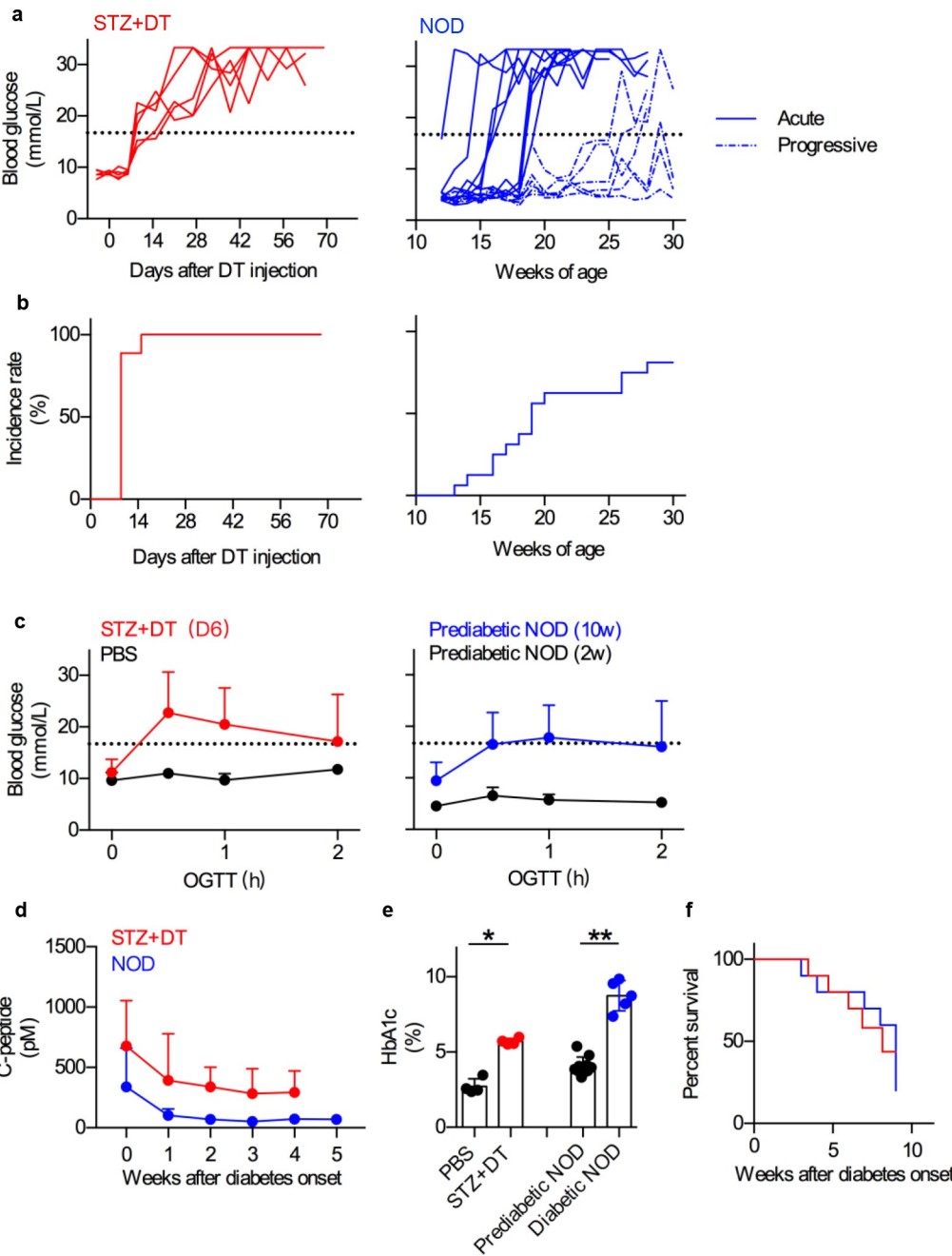

**Fig 2. Diabetes symptoms covered by both STZ+DT and NOD model. a**. Blood glucose fluctuations of STZ+DT mice (left, red, n = 5) and NOD mice (right, blue, n = 15). Blood glucose over 16.7 mmol/L was identified as hyperglycemia (Black dash line). Acute diabetic NOD mice (Blue lines) and progressive ones (Blue dash lines) were categorized. **b**. Diabetes incidence rate of STZ+DT mice (left, n = 9) and NOD mice (right, n = 16). **c**. abnormal OGTT of prediabetic STZ+DT mice at day 6 (left) and prediabetic NOD mice at 2-week or 10-week of age (right). Statistics comparing to PBS control or pre-diabetic mice were shown. n = 5. N = 3. **d**. Serum C-peptide loss after diabetes onset. Statistics comparing to week 0 were shown. n = 5. N = 3. **e**. Hemoglobin (HbA1c) of STZ+DT mice (8 weeks after diabetes onset) and NOD mice (10-week for pre-diabetic and 17-week for diabetic). n = 5. **f**. Survival of STZ+DT and NOD mice after diabetes onset. n = 10. *, $p < 0.05$; **, $p < 0.01$.

hemoglobin (HbA1c) [21] was also increased in STZ+DT mice (**Fig 2E**). In the end, survival of diabetic mice showed similar result in both models, with 50% morbidity around 8 weeks, suggesting that although pre-diabetic pathogenesis was faster in STZ+DT model than NOD, similar diabetic complications were developed after disease onset (**Fig 2F**). Collectively, these results suggested that the double-hit STZ+DT model developed conventional diabetic symptoms depicted by NOD mice, with a relatively controlled onset and much increased efficiency.

## Autoimmune insulitis in STZ+DT model

Various deviations in disease development of NOD mice from patients have been reported. Among them, one crucial aspect of diametrical contrast is in T cell infiltration bias. Cytolytic CD8[+] T cells predominantly kill β cells in islet of T1D patients [22], which is likely the most critical pathogenesis leading to diabetes. Although CD8 T cells were required for pathogenesis in NOD, CD4[+] T helper cell infiltration is robust and dominant [8]. To investigate if the double-hit model represents a more precise simulation of human disease, inflammatory pancreas infiltrates were analyzed. Lymphocyte accumulation and islet destruction were identified in pre-diabetic STZ+DT pancreas, similar to in NOD mice at 10-weeks (**Fig 3A and 3B**). Among CD3[+] T cells accumulated (**Fig 3C**), CD8[+] T cells dominated in STZ+DT pancreas significantly more than CD4[+] T cells (**Fig 3D and 3E**). Whereas in NOD mice, both CD8[+] and CD4[+] T cells showed similar rate of infiltration (**Fig 3E**, $p = 0.1$). Equally demontrating its similarity to human disease, anti CD8 antibody almost completely prevented diabetogenesis in the STZ +DT model (**Fig 3F**). To further identify auto-antigen reactive CD8 T cells in STZ+DT model, infiltrated CD8[+] T cells were isolated and re-stimulated with β-cell epitopes *in vitro*. $GAD65_{114-122}$ or $IAPP_{5-13}$ (prepro) triggered IFN-γ responses in STZ+DT mice, comparing to non-relevant epitope control $OVA_{257-264}$ (**Fig 3G**). Thus, autoimmune CD8 T cells were induced and play a significant role in STZ+DT model.

A clear onset of diabetic ketoacidosis (DKA) is a hallmark of the human disease [9]. The transition is less evident in NOD mice. In double-hit model, plasma β-hydroxybutyrate acid (BHA, indicator for ketoacidosis) was significantly higher than PBS control group (**Fig 3H**). In comparison, NOD mice showed consistent level of BHA from pre-diabetic (2-week) to diabetic phase, indicating disrupted glucose metabolism early-on, unlike human disease progression (**Fig 3H**). Overall, this double-hit model showed a set of clinical findings unattainable in the NOD system, and replicated pivotal features of CD8[+] T cell dominance, epitope analogy to human and diabetic ketoacidosis (**Fig 3I**).

## Tuning disease progression by each hit

Finally, we sought to control the disease progression through tuning each hit. Severity of pancreas damage (1[st] hit) was tuned by STZ dosage (5*STZ, 3*STZ-the standard protocol, 2*STZ and 1*STZ) (**Fig 4A**, left panel). As expected, pancreas damage severity correlated with disease onset (**Fig 4A**, right panel) and survival (**Fig 4B**). Autoimmune activation (2[nd] hit) was dictated by DT depletion of Tregs. Heterozygous DEREG mice spontaneously develop DTR minus Tregs, due to stochastic genomic silencing of BAC GFP-DTR transgene [23], which provided an angle for Treg depletion tuning. We confirmed this notion in our double hit model (**Fig 4C**). In our assays, these heterozygous mice showed limited depletion of endogenous Treg (**Fig 4D**) and decelerated diabetes progression (**Fig 4E**). To summarize, 1[st] hit manipulation by STZ dosage in homozygous mice from 3*STZ to 5*STZ accelerated disease onset from day7~9 to day 5~8 (**Fig 4F**, magenta versus red), and 2[nd] hit manipulation by DEREG heterogeneity delayed the disease from day 7~9 to day 15~35 (**Fig 4F**, red versus brown). Regarding the batch effect in the timing of disease onset, we found that similar to the

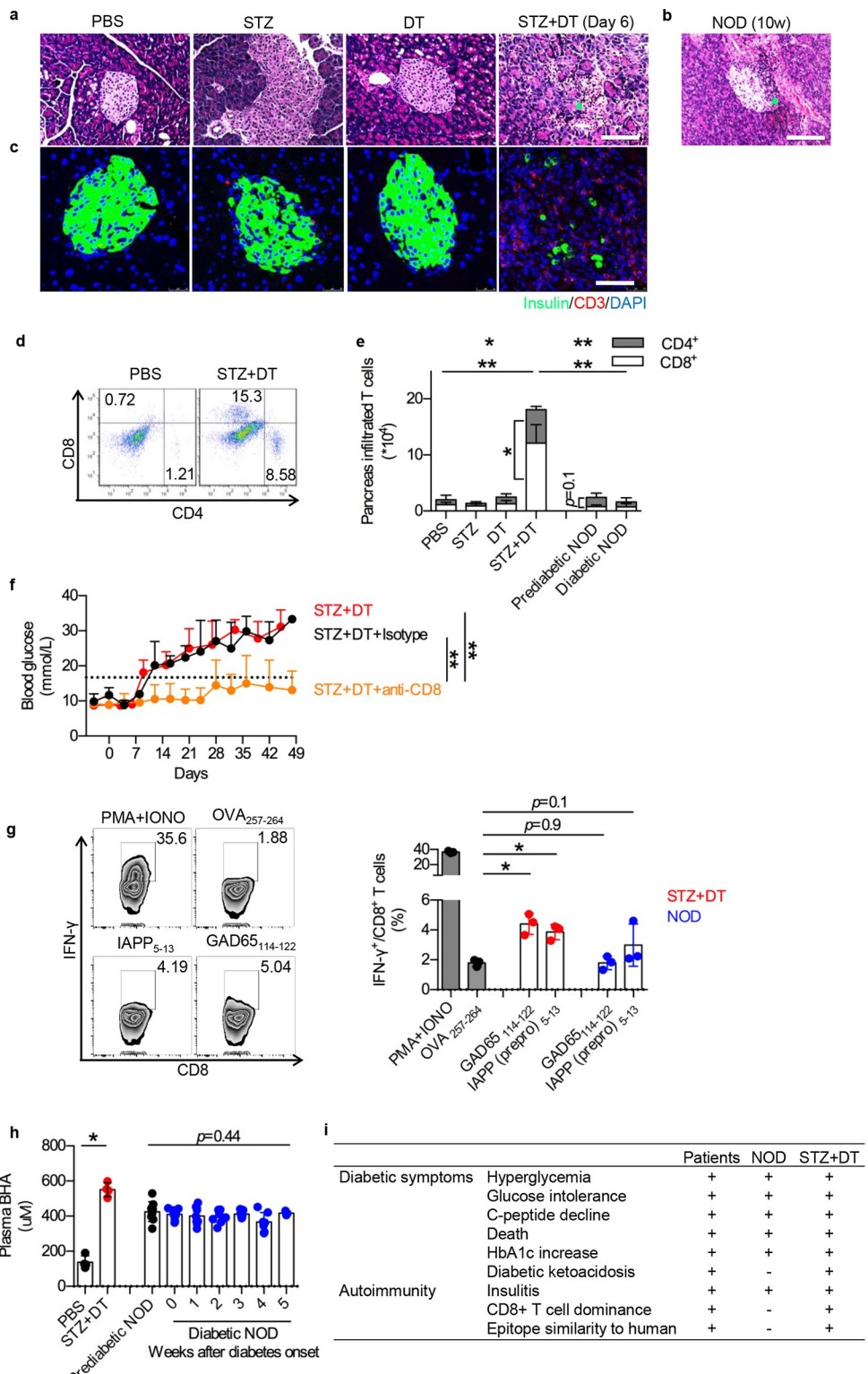

**Fig 3. CD8+ T cell infiltration and severe ketoacidosis in STZ+DT mice. a-b**. H&E staining of islets from STZ+DT mice and control groups at day 6 (**a**), and prediabetic NOD mice at 10-week old (**b**). Arrow pointed at inflammatory infiltrates. **c**. Immunofluorescence of CD3 (red), insulin (green) and DAPI (blue) of islets. Scale bars represent 50μm. **d**. T cells subsets in pancreatic infiltrates. CD8+ and CD4+ T cells were identified by flow cytometry. Mice age as in 3a and 3b. **e**. Statistical analysis of 3d. Difference was calculated between STZ+DT and control groups (top left, horizontal

comparisons) or between STZ+DT and NOD model (top right, horizontal comparisons). Both CD4$^+$ T cells (gray) and CD8$^+$ T cells (white) were analyzed. Statistical significance was also calculated with CD8$^+$ versus CD4$^+$ T cells within STZ+DT or prediabetic NOD (vertical comparisons). n = 5. N = 3. **f**. Blood glucose fluctuation after CD8$^+$ T cell depletion. Anti-CD8 neutralizing antibody (Orange) or isotype control (Black) were *i.p.* injected at day 3, 6, 9, 12. n = 4. N = 3. **g**. Auto-antigen specific IFN-γ response in CD8$^+$ infiltrates. IFN-γ response in CD8$^+$ infiltrates in STZ+DT mice (left) and statistical analysis (right). PMA+Ionomycin (PAM+IONO) was added as positive control. OVA$_{257-264}$ was added as negative control. GAD65$_{114-122}$ and IAPP$_{5-13}$ stimulated responses were analyzed. Mice age as in 3a and 3b. n = 3. N = 3. **h**. Ketoacidosis in STZ+DT mice (Day 30) and NOD mice (2-week old for pre-diabetic and after diabetes onset). Plasma β-hydroxybutyric acid (BHA) was analyzed. n = 5. N = 3. $^*$, $p < 0.05$. **i**. Similarities of STZ+DT model comparing to patients and NOD mice.

original design shown in Fig 2, the difference was minimal (less than one week) among the batches even upon these detailed tuning of experimental protocols (**Fig 4F**), in strong contrast to the dispersed diabetes occurrences in 10s of weeks of NOD mice (**Fig 4G**).

## Discussion

While existing type I diabetes models are instrumental in our understanding of this disease, they were not generated in accordance with modern immunological knowledge on this autoimmunity. This to a great extent limits our ability to generalize findings obtained in these animals to the human disease. The lack of clinical fruition based on the "NOD" diabetes research lends credence to this framework deficiency. Here we put the 'double-hit' hypothesis to test and created a novel animal model of autoimmune diabetes. This model was confirmed with long-time hyperglycemia, C-peptide deficiency and elevated HbA1c, accompanied by DKA. More importantly, it mainly promoted T cells infiltration and subsequent activation of cytolytic CD8$^+$ T cells. Consequently, islet destruction led to insulin-deficient type 1 diabetes. The speed and severity of the disease development were under control via tuning STZ schedules or genotypes of DEREG mice.

The merits of spontaneous model are irreplaceable, such as inheritable genetic traits and broad disease diversities. From another direction, the double hit proposal comes from more refined knowledge of human autoimmune diabetes. Autoimmunity is complex by nature, likely more so for laboratory animal models mimicking it. Humans are exposed to environmental factors, such as ingested chemicals or Coxsackie virus infections. Those confirmed contributing factors of type 1 diabetes are clearly lacking in existing models [1, 24]. On the other hand, immune preferences were also supported by genetic studies [25]. However, neither factor sufficiently leads to disease onset. A subtle disturbance of the immune system likely also predates the disease. This dilemma spurred us to look for a niche for both, specifically a temporal-spatial alignment of both environmental trigger and immuno-defects. In fact, this double hit proposal may be relevant to other autoimmune diseases, which certainly requires experimental verification in the future. In this report, we found multiple low-dose STZ and DEREG system as tools to fulfill this alignment in mouse. By this rational and simple reconstruction of autoimmune diabetes, we believe that the resulting protocol can competently replace or be used in parallel to the current T1D models, with outcomes at least in some aspects more analogous to the human disease than what to be expected currently. In our view, the genetic predisposition would likely be better revealed with the NOD system, our model excels in progression simulation. It is up to the researchers to determine which model is better suited for their specific needs.

## Methods

### Mice

DEREG (FOXP3-DTR/EGFP) mice were obtained from Jackson Lab (Bar Harbor, ME, USA) then bred into Balb/c background, and in all experiments this Balb/c DEREG mice were used.

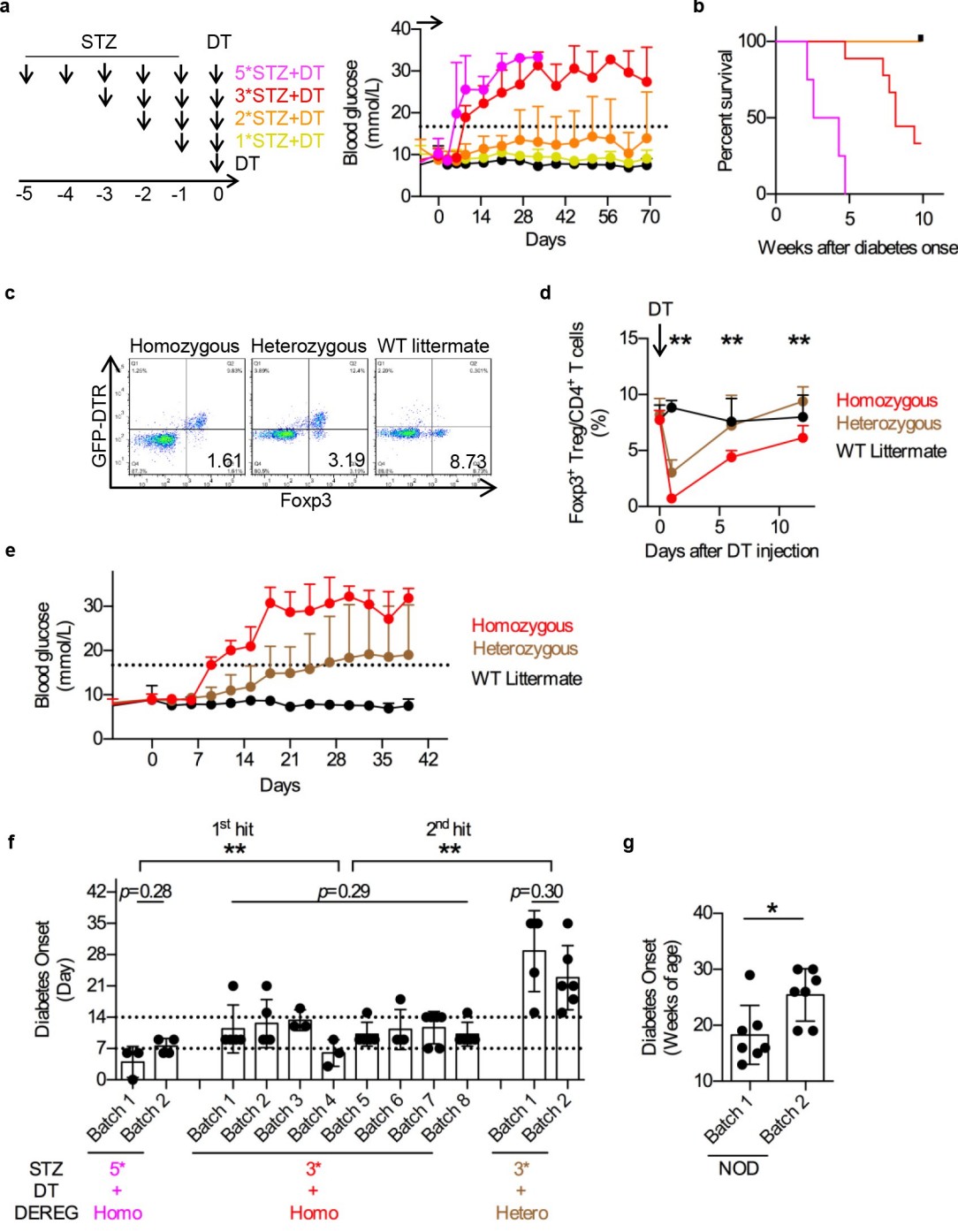

**Fig 4. Manipulations of each hit tuned diabetes progression. a**. Pancreas damage severity affected diabetes progression. Left, multiple low-dose STZ schedules. In addition to standard 3*STZ+DT schedule (Red), stronger (5*STZ+DT, magenta) or weaker (2*STZ+DT, orange, or 1*STZ+DT, yellow) STZ damage schedules were analyzed. Right, blood glucose fluctuation after model setup. DT was given as control (Black). Arrow represents model setup from day -5 to day 0. Dash line represents blood glucose at 16.7 mmol/L. n = 5. N = 3. **b**. Survival after diabetes onset. Color codes as in 4a. n = 5. **c-e**. Autoimmune severity affected diabetes progression. **c**. Genotype of homozygous and heterozygous DEREG mice. Wildtype littermates were added as negative control. Numbers showed Foxp3+GFP-DTR− Treg percentage in CD4+ spleno-T cells, analyzed by intracellular staining. n = 3. N = 3. **d.** Foxp3+ Treg dynamic after DT depletion of DTR+ Tregs. Statistical significance was analyzed between homozygous (Red) and heterozygous (Brown) groups at day 1, 6 and 12. n = 6. N = 3. **e**. Blood glucose fluctuation after model setup in homozygous or heterozygous DEREG mice. Blood glucose over 16.7 mmol/L was identified as hyperglycemia (Dash line). Statistics between Homozygous and Heterozygous mice were shown for each testing day. n = 6. N = 3. **f** Disease onset date of 1st hit or 2nd hit

manipulations. 1st hit manipulation was compared between 5*STZ+DT (magenta) and 3*STZ+DT (red) groups, both in homozygous DEREG mice. 2nd hit manipulations was compared between homozygous (Red) and heterozygous (Brown) DEREG mice, both under 3*STZ+DT standard schedule. Batch effects were also calculated within the same schedule. n = 4~6. **g**. Disease onset date of two batches of NOD mice. n = 7. *, $p < 0.05$; **, $p < 0.01$.

Comparing to original C57BL/6 DEREG mice, Balb/c DEREG was chosen because of robustness (lower variation between mice and batches) to multiple low dose STZ, which may be due to biased chemical responses between two strains [26]. Treg depletion efficiencies were similar between C57BL/6 and Balb/c DEREG mice. Non-obese diabetic (NOD.shilt/J) mice were obtained from Jackson Lab and female NOD mice were used in this study. Mice were kept in specific pathogen-free condition with 12hr light cycle at Fudan University Shanghai Medical College. All experiment protocols were approved ahead by Animal Ethics Committee of School of Basic Medical Sciences at Fudan University (Approval number: 20180302–106) and performed in compliance with guidelines provided by Institutional Animal Care and Use Committee (IACUC) and ARRIVE.

## Survival study

Survival studies were conducted in 10 NOD mice and 38 DEREG mice injection with various dose of STZ combination and a single dose of DT up to 69 days. 24 of all mice were euthanized and identified as weight loss more than 20%, lack of mobility, weak breathing, dehydration, when they reached the humane endpoints, which were monitored daily. All efforts were taken to minimize the suffering and distress, including cleaning the cage and changing the packing once 1/3 of the cage was wet due to urorrhagia of mice, and replenishing the water in case of polydipsia of the mice, which were attributed to symptoms of diabetes. In addition, the mice were monitored reaching the humane endpoint and no improvement within 8 hours were euthanized via CO2 inhalation and subsequent cervical dislocation. No special training in animal care or handling was required.

## Double-hit model and diabetes

DEREG mice were randomly assigned into groups. At the days of multiple low-dose STZ injection, streptozotocin (STZ; Sigma-Aldrich, St. Louis, MO) was freshly prepared in citrate buffer (pH = 4.5). Within fifteen minutes after dissolving, STZ was *i.p.* injected at 50mg/kg body weight dosage. To deplete Treg, diphtheria toxin (DT; Sigma-Aldrich, St. Louis, MO) was dissolved in PBS (pH = 7.4) and mice were *i.p.* injected with 0.25μg DT. Unless stated otherwise, double-hit model mice were setup with three-day injections of low-dose STZ (Day -3, -2, -1) followed by a single DT injection (Day 0) throughout the study. Blood glucose was monitored (JPS5-7, Yichen Co, Ltd, Bejing) and diabetic hyperglycemia was identified above 16.7 mmol/L blood glucose. Oral glucose tolerance tests (OGTT) were performed on 6-hour fasted mice with 2g/kg glucose dispersion in water via oral gauge, followed by blood glucose monitoring at 0, 30, 60 and 120 minutes. Serum C-peptide from fasted mice was measured by ELISA (ALPCO, America). Hemoglobin A1c was measured in venous blood via ELISA on day 60 (Crystal Chem Co. Ltd, Japan). Plasma β-hydroxybutyric acid (BHA) was detected on day 30 (EK-Bioscience, Switzerland).

## Histology and immunofluorescence (IF)

Pancreas were fixed in 4% Paraformaldehyde (PFA) then 4μm paraffin sections were prepared. Hematoxylin-eosin (H&E) staining and immunostaining were prepared for macroscopy

analysis. For IF, following primary antibodies were used: Rabbit anti-mouse insulin (1:100, Abways, CY5471), Rabbit polyclonal anti-mouse Glucagon (1:100, Servicebio, gb11097), Rabbit monoclonal anti-mouse CD3 (1:100, Abcam, ab16669). IF images were acquired under confocal microscopy (Leica, Germany). Insulin positive β cells were identified using Image J Software (NIH, USA).

### Pancreatic infiltrated T cell isolation and re-stimulation

At day 6, the pancreas from STZ+DT-treated mice or PBS control mice were grinded in 1ml RPMI 1640 and pressed through a 70-μm cell strainer, centrifuged and resuspended. Density gradient centrifugation was performed with Ficoll (Dakewe Biotech Co., Ltd., Shenzhen, China). Ficoll enriched lymphocytes were washed, centrifuged and resuspended in RPMI 1640 with 10% FBS (Biological Industries, USA). $10^5$ lymphocytes were re-stimulated with $CD8^+$ T cell epitopes in presents of mitomycin treated spleno-DCs: $GAD65_{114\text{-}122}$ (VMNILLQYV) (10μg/ml) and IAPP (prepro) $_{5-13}$ (KLQVFLIVL) (10μg/ml). PMA (0.1μg/ml) and ionomycin (1μg/ml) were added as positive control. $OVA_{257\text{-}263}$ (SIINFEK) was used as negative control respectively [27]. 1× brefeldin A (eBioscience) was added into culture system for 6h to block cytokine secretion. All islet peptides were synthesized (Sangon, Shanghai, China). Then cells were stained and required using Flow Cytometry.

### Flow cytometry

Cells were suspended in FACS buffer (2% FBS in PBS) and pre-stained for the surface markers with anti-CD4 (clone GK1.5) and anti-CD8 (clone 53–6.7) (Biolegend). For intracellular staining, cells were then fixed for 1 hour at 4˚C with Foxp3/transcription factor staining buffer set (eBioscience, San Diego, CA), then stained for 1 hour at room temperature with anti-Interferon-γ (clone XMG1.2, eBioscience) or anti-Foxp3 (clone FJK-16s, eBioscience). Data were acquired on Fortessa flow cytometer (BD Biosciences, San Diego, CA, USA) then analyzed by FlowJo (Tree Star, Ashland, OR, USA).

### CD8+ T cell depletion *in vivo*

Anti-mouse CD8a neutralizing antibody (clone 2.43, InVivoMab) was purchased from BioX-cell. Mice were *i.p.* injected with 200μg neutralizing antibody or 200μg isotype on day 3, 6, 9 and 12 post DT injection.

### Statistical analysis

Data were analyzed by Graphpad Prism (San Diego, CA) and demonstrated as mean±SEM. Mann-Whitney t-test or Two-way ANOVA was used. *, $p<0.05$. **, $p<0.01$.

### Supporting information

**S1 File.**
(ZIP)

### Acknowledgments

We thank Ms. Yiwei Zhong at Shanghai Medical College of Fudan University her technical assistant during this study, and Mr. Zhonghuai He and Ms. Yue Peng at Advaccine Biotech for their technical guidance and assistance in this work. We appreciate Dr. Yan Shi for his critical review.

## Author Contributions

**Conceptualization:** Bin Wang.

**Data curation:** Xiang Jin.

**Formal analysis:** Fan Yu, Shushu Zhao.

**Investigation:** Fan Yu, Xian Zhou.

**Resources:** Gan Zhao.

**Validation:** Sheng Jiang.

**Writing – review & editing:** Shuang Geng.

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
