## [Decision Letter · Decision Letter 0]

6 Sep 2021

PONE-D-21-15861Rational Construction of Controllable Autoimmune Diabetes Model Depicting Clinical FeaturesPLOS ONE

Dear Dr. Wang,

Thank you for submitting your manuscript to PLOS ONE. After careful consideration, we feel that it has merit but does not fully meet PLOS ONE’s publication criteria as it currently stands. Therefore, we invite you to submit a revised version of the manuscript that addresses the point raised below my signature in relation to the in vivo animal experiments.

We look forward to receiving your revised manuscript.

Kind regards,

Paolo Fiorina, MD, PhD

Academic Editor

PLOS ONE

****Request from staff editors****

We note that your study design may include death of a regulated animal as a likely outcome or planned experimental endpoint. At this time, we request that you please report additional details in your Methods section regarding animal care and use for the survival study, as per our editorial guidelines (http://journals.plos.org/plosone/s/submission-guidelines#loc-humane-endpoints). 

For easy reference, we have attached a checklist that may be relevant for your submission. Please complete all items on the checklist at the following link: http://journals.plos.org/plosone/s/file?id=bb1d/plos-one-humane-endpoints-checklist.docx .Please upload the completed checklist as file type “Other” when resubmitting your manuscript. This document is for internal journal use only and will not be published if your article is accepted. We very much appreciate your attention to these requests and support of improved reporting standards in PLOS ONE submissions.

Journal Requirements:

Reviewers' comments:

Reviewer's Responses to Questions

**Comments to the Author**

1. Is the manuscript technically sound, and do the data support the conclusions?

Reviewer #1: Yes

2. Has the statistical analysis been performed appropriately and rigorously? 

Reviewer #1: Yes

3. Have the authors made all data underlying the findings in their manuscript fully available?

Reviewer #1: Yes

4. Is the manuscript presented in an intelligible fashion and written in standard English?

Reviewer #1: Yes

5. Review Comments to the Author

Reviewer #1: This is an interesting study and the authors have collected good data. The paper is generally well written and structured. Results are clear and well discussed in the manuscript. Supplementary materials are useful and help to keep the study understandable.

6. PLOS authors have the option to publish the peer review history of their article (what does this mean?). If published, this will include your full peer review and any attached files.

Reviewer #1: **Yes: **Lara Loreggian

---

## [Author Response · Author response to Decision Letter 0]

9 Sep 2021

Point 1: ****Request from staff editors****

We note that your study design may include death of a regulated animal as a likely outcome or planned experimental endpoint. At this time, we request that you please report additional details in your Methods section regarding animal care and use for the survival study, as per our editorial guidelines (http://journals.plos.org/plosone/s/submission-guidelines#loc-humane-endpoints). 

Response 1: Thanks for your request, we have added the paragraph to describe animal care and use for the survival study in our Methods section, from line 198 to line 208, including description of the humane endpoints in this study.

Point 2: Journal Requirements:

Response 2: Thank you for the suggestion. I have checked the whole reference listed online, no any article cited was retracted. No necessary change for the current reference list.

---

## [Decision Letter · Decision Letter 1]

3 Nov 2021

Rational Construction of Controllable Autoimmune Diabetes Model Depicting Clinical Features

PONE-D-21-15861R1

Dear Dr. Wang,

We’re pleased to inform you that your manuscript has been judged scientifically suitable for publication and will be formally accepted for publication once it meets all outstanding technical requirements.

Kind regards,

Paolo Fiorina, MD, PhD

Academic Editor

PLOS ONE

Additional Editor Comments (optional):

Reviewers' comments:

Reviewer's Responses to Questions

**Comments to the Author**

1. If the authors have adequately addressed your comments raised in a previous round of review and you feel that this manuscript is now acceptable for publication, you may indicate that here to bypass the “Comments to the Author” section, enter your conflict of interest statement in the “Confidential to Editor” section, and submit your "Accept" recommendation.

Reviewer #1: All comments have been addressed

2. Is the manuscript technically sound, and do the data support the conclusions?

Reviewer #1: Yes

3. Has the statistical analysis been performed appropriately and rigorously? 

Reviewer #1: Yes

4. Have the authors made all data underlying the findings in their manuscript fully available?

Reviewer #1: Yes

5. Is the manuscript presented in an intelligible fashion and written in standard English?

Reviewer #1: Yes

6. Review Comments to the Author

Reviewer #1: (No Response)

7. PLOS authors have the option to publish the peer review history of their article (what does this mean?). If published, this will include your full peer review and any attached files.

Reviewer #1: No

---

## [Editor Report · Acceptance letter]

30 Dec 2021

PONE-D-21-15861R1 

Rational Construction of Controllable Autoimmune Diabetes Model Depicting Clinical Features 

Dear Dr. Wang:

I'm pleased to inform you that your manuscript has been deemed suitable for publication in PLOS ONE. Congratulations! Your manuscript is now with our production department. 

Kind regards, 

on behalf of

Dr. Paolo Fiorina 

Academic Editor

PLOS ONE